# Reduced Exertion High-Intensity Interval Training is More Effective at Improving Cardiorespiratory Fitness and Cardiometabolic Health than Traditional Moderate-Intensity Continuous Training

**DOI:** 10.3390/ijerph16030483

**Published:** 2019-02-07

**Authors:** Tom F. Cuddy, Joyce S. Ramos, Lance C. Dalleck

**Affiliations:** 1Recreation, Exercise & Sport Science Department, Western Colorado University, Gunnison, CO 81231, USA; tcuddy@lonepeakpt.com; 2SHAPE Research Centre, Exercise Science and Clinical Exercise Physiology, College of Nursing and Health Sciences, Flinders University, Bedford Park, SA 5042, Australia; joyce.ramos@flinders.edu.au; 3Centre for Research on Exercise, Physical Activity and Health, School of Human Movement and Nutrition Sciences, The University of Queensland, Brisbane, QLD 4072, Australia

**Keywords:** cardiometabolic risk factor, metabolic syndrome, sprint interval training, translational research

## Abstract

This study sought to determine the effectiveness of an 8 wk reduced-exertion high-intensity interval training (REHIT) at improving cardiorespiratory fitness (CRF) and positively modifying cardiometabolic health in the workplace environment. Participants (n = 32) were randomized to two groups: (1) One group (n = 16) was prescribed an 8 wk REHIT program, and (2) one group (n = 16) was prescribed moderate-intensity continuous training (MICT). Cardiometabolic risk factors and CRF were measured at baseline and 8 wks. After 8 wks, changes in CRF (REHIT, 12%; MICT, 7%), systolic blood pressure (REHIT, −5%; MICT, −2%), waist circumference (REHIT, −1.4%; MICT, −0.3%), and metabolic syndrome (MetS) severity (MetS z-score: REHIT, −62%; MICT, 27%) were more favorable (*p* < 0.05) in the REHIT group relative to the MICT group. Interestingly, there was a significantly greater proportion of participants in the REHIT group (75%, 9/12) who had a favorable change in the MetS z-score (Δ > −0.60) relative to the MICT group (47%, 7/15). The main finding of the present study is that 8 wks REHIT elicited more potent and time-efficient improvements in CRF and cardiometabolic health when compared to traditional MICT. This study provides critical evidence for implementation of the sprint interval training (SIT) paradigm from the scientific literature into a real-world workplace setting.

## 1. Introduction

Findings from numerous epidemiological studies clearly demonstrate that physical inactivity is associated with a higher prevalence of most cardiovascular disease (CVD) risk factors, including abnormal lipids, high blood pressure, metabolic syndrome, obesity, and type 2 diabetes [1]. Additionally, physical inactivity is linked with increased risk of certain forms of cancers, poor psychological health, and an overall diminished quality of life [2]. Moreover, research findings also exhibit a robust inverse relationship between physical activity levels and risk of mortality from CVD and all-causes. Last, it has been estimated that physical inactivity contributes annually to approximately 250,000 premature deaths [3]. However, despite its widespread benefits, engagement in physical activities or exercise remains scarce, primarily reported to be due to a ‘lack of time’ [4]. 

Over the past decade, the concept of high-intensity interval training (HIIT) has captivated the attention of the scientific community due to its superior ability to improve cardiorespiratory fitness [5] and cardiometabolic health [6,7,8] for a lesser weekly time-commitment relative to the current exercise guidelines [2] of moderate-intensity continuous training (MICT). HIIT involves multiple (~4–10 repetitions) brief bouts (20 s–5 min) of high-intensity exercise (80%–100% peak heart rate (HRpeak)), interspersed with either rest or low-intensity workloads throughout an exercise session. Another prominent form of interval training is known as sprint interval training (SIT), which is characterized by repeated (6–10 bouts) 20–30 s all-out supramaximal sprints (>100% VO_2_max). This strategy enables unfit individuals to accumulate periods of vigorous to high-intensity exercise that would otherwise not be possible if executed continuously. However, one drawback to the protocols employed in the majority of previous HIIT/SIT studies [9] is that they were not actually time-efficient, with most HIIT/SIT protocols requiring a time commitment (~120 min/week) that is similar to the current recommended exercise guideline of 150 min per week of MICT. Moreover, it has also been suggested that the potential for a negative perceptual response to high-intensity exercise heightens with increasing repetition [10]. Thus, for HIIT or SIT to be a feasible option to improve public health, it must be time-efficient by specifically reducing the number of bouts (intervals) performed at vigorous to high-intensity exercise.

Recent research has demonstrated that a modified version of SIT known as reduced exertion high intensity training (REHIT), characterized by minimal sprint durations and repetitions (2 × 20 s sprints), elicits substantial cardiometabolic health benefits (i.e., CRF and glycemic control improvements) in a more time-efficient fashion relative to traditional HIIT [11,12]. However, despite the reduced exercise volume evident in this REHIT protocol, its safety still remains questionable given its ‘all-out’ nature. It has therefore recently been suggested that additional future research is warranted to identify safe and effective REHIT protocols [9]. It also remains unknown how effectively the REHIT paradigm can be translated to a real-world workplace setting. Accordingly, we sought (1) to quantify the acute cardiovascular and metabolic responses to REHIT, and (2) to determine the effectiveness of 8 wk chronic REHIT at improving CRF and positively modifying cardiometabolic health in the workplace environment. It was hypothesized that: (1) An acute bout of REHIT would be a safe and effective form of SIT, and (2) chronic REHIT would lead to time-efficient improvements in cardiorespiratory fitness and cardiometabolic health. 

## 2. Materials and Methods 

A cohort (n = 32) of men (n = 16) and women (n = 16) were recruited from the faculty/employee population of a local university and hospital via advertisement through the university website, local community newspaper, and through word-of-mouth. Participants were eligible for inclusion into the study if they were low risk-to-moderate risk and physically inactive as defined by the American College of Sports Medicine [13]. Exclusionary criteria included evidence of cardiovascular pulmonary and/or metabolic disease. This study was approved by the Human Research Committee at Western State Colorado University. Prior to participation, each participant signed an informed consent form. 

### 2.1. Chronic Responses (8 wk) to the REHIT vs. MICT

After recruitment, participants were randomized by age and sex to one of two exercise-training groups and subsequently completed baseline testing (Figure 1). Participants performed 8 wks of supervised exercise training according to one of two programs:One group (n = 16) was prescribed an 8 wk REHIT program consisting of intense ride options performed on CAR.O.L ™. All REHIT were performed in two local workplace environments (university and hospital). One CAR.O.L ™ cycle ergometer unit was placed in each location in order for participants to perform REHIT in the workplace environment.One group (n = 16) was prescribed 8 wk standardized MICT (5 days/wk of 30 min aerobic exercise at 50%–65% heart rate reserve—HRR).

At baseline and post-program, participants performed a graded exercise test on a cycle ergometer to determine the maximal heart rate and maximal CRF. Resting blood pressure, body composition, fasting blood lipids and blood glucose, waist circumference, and weight were assessed in order at baseline and post-program, at 48–72 h following the last training session. Collectively, these measures were obtained to compare the effectiveness of an 8 wk REHIT vs. MICT exercise training for improving cardiometabolic health. The experimental flow diagram and week-to-week exercise prescription for the days/times of training are presented in Figure 1.

### 2.2. Acute Responses to REHIT

To quantify the acute cardiovascular and metabolic responses to REHIT, an Oxycon Mobile portable calorimetric measurement system and Polar F1 heart rate monitor were worn by each participant throughout two REHIT sessions performed on CAR.O.L™, a cycle ergometer unit developed by Integrated Health Partners (London, UK). The CAR.O.L ™ REHIT protocol utilises self-learning algorithms called Cardiovascular Optimisation Logic to adapt the resistance level according to participant’s weight, power output and fatigue index in every session as they get progressively fitter and stronger. Details for the REHIT ride options are as follows: Intense ride: A 10 min REHIT ride comprised of a 3 min warmup, 20 s sprint, 3 min recovery (slow pedal), 20 s sprint, and 3 min cooldown.

The acute responses to the two REHIT were measured (and averaged) during week 4–5 of the intervention, with these testing sessions also serving as two of the actual training sessions.

### 2.3. Anthropometric Measurements

Participants were weighed to the nearest 0.1 kg on a medical grade scale and measured for height to the nearest 0.5 cm using a stadiometer. Percentage of body fat was determined via skinfolds [13]. Skinfold thickness was measured to the nearest ± 0.5 mm using a Lange caliper. All measurements were taken on the right side of the body using standardized anatomical sites (three-site) for men and women. These measurements were performed until two were within 10% of each other. Waist circumference measurements were obtained using a cloth tape measure with a spring loaded-handle. A horizontal measurement was taken at the narrowest point of the torso (below the xiphoid process and above the umbilicus). These measurements were taken until two were within 0.5 mm of each other.

### 2.4. Fasting Blood Lipid and Blood Glucose Measurement

A fasting blood sample was collected and analyzed for measurement of lipids and glucose. Participants’ skin was punctured using lancets and a fingerstick sample was collected into a heparin-coated 40 μL capillary tube. Blood was allowed to flow freely from the fingerstick into the capillary tube without milking of the finger. Samples were analysed via a Cholestech LDX System according to strict standardized operating procedures. The LDX Cholestech measured the total cholesterol, high density lipoprotein (HDL) cholesterol, low density lipoprotein (LDL) cholesterol, triglycerides, and blood glucose in the fingerstick blood. A daily optics check was performed on the LDX Cholestech analyzer used for the study. 

### 2.5. Resting Blood Pressure Measurement

The procedure for the assessment of resting blood pressure, outlined elsewhere, was followed [13]. Briefly, participants were seated quietly for 5 min in a chair with a back support with the feet on the floor and the arm supported at the heart level. The left arm brachial artery systolic and diastolic blood pressure were measured using a sphygmomanometer in duplicate and separated by 60 s. The means of the two measurements were reported for the baseline and post-program values. 

### 2.6. Metabolic Syndrome z-Score

A continuous risk score assessment scale (MetS z-score) has been used previously to identify changes in metabolic syndrome (MetS) risk factors following MICT and HIIT interventions [14]. The MetS severity was presented as a sex-specific MetS z-score calculated using the following equations [15]:
MetS z-score_men_ = [(40 − HDL-C)/8 × 9] + [(TG − 150/69)] + [(FG − 100)/17 × 8] + [(WC − 102)/11 × 5] + [(MAP− 100)/10 × 1]; (1)
MetS z-score_women_ = [(50 − HDL-C)/14 × 5] + [(TG − 150/69)] + [(FG − 100)/17 × 8] + [(WC − 88)/12 × 5] + [(MAP − 100)/10 × 1], (2)
where FG = fasting glucose; HDL-C = high-density lipoprotein cholesterol; MAP = mean arterial pressure; TG = triglycerides; and WC = waist circumference.

### 2.7. Maximal Exercise Test

Participants were prepared for maximal exercise on the cycle ergometer (Viasprint 150P; Sensormedics Corp., Palm Springs, CA, USA) during which gas exchange data (Oxycon Mobile system) and HR (Polar Electro, Woodbury, NY, USA) were assessed. Participants completed 2 min of pedaling at 50 W as a warm-up. During exercise, power output was increased in a step like manner equal to 5 W/30 s for women and 5 W/20 s for men to elicit volitional fatigue in approximately 7–11 min. Pedal cadence was maintained at 70–90 rev/min, with volitional fatigue representing a failure to sustain a pedal cadence greater than 40 rev/min. The criteria for the attainment of maximal CRF were two out of three of the following: (1) A plateau (ΔVO_2_
< 150 mL/min) in VO_2_ with increases in workload, (2) maximal respiratory exchange ratio (RER) > 1.1, and (3) maximal HR within 15 beats/min of the age-predicted maximum (220—age). 

### 2.8. Testing Sessions to Quantify Acute Cardiovascular and Metabolic Responses to REHIT

Prior to the start of the testing sessions, both the heart rate monitor and portable metabolic analyzer were attached to the participant. Participants were also familiarized with the breathing apparatus and provided an explanation of the testing instructions and precautions. Participants subsequently performed two REHIT testing sessions (i.e., the intense ride) on separate days. At the first testing session, after being connected to the Oxycon Mobile system and Polar F1 heart rate monitor, participants rested quietly for 5 min in a seated position. The last minute of breath-by-breath and heart rate (HR) data were averaged and considered to be the resting metabolic rate (VO_2_) and resting HR.

### 2.9. Exercise Intensity and Metabolic Calculations for Acute Responses

The individual heart rate reserve (HRR) was determined as the difference between the resting and HRmax values. Percent HRR was calculated by subtracting resting HR from the REHIT HR responses, dividing by HRR, and then multiplying the quotient by 100. Likewise, individual oxygen uptake reserve (VO_2_R) was quantified by taking the difference between individual resting VO_2_ and maximum VO_2_ values. Percent VO_2_R was calculated by subtracting resting VO_2_ from the REHIT VO_2_ responses, dividing by VO_2_R, and then multiplying the quotient by 100. The metabolic equivalent (MET) for REHIT was determined by dividing the REHIT VO_2_ by the individual resting VO_2_ value (i.e., 3.5 mL/kg/min). Energy expenditure (kcal/min) for the REHIT sessions were calculated by multiplying the above-calculated MET equivalent of the REHIT sessions by the individual body mass, dividing by 1000, and multiplying by the caloric equivalent for the respiratory exchange ratio or RER. An RER of 1.0 was assumed and equates to an energy cost of ~5.0 kcal/L oxygen. Means for each exercise intensity and metabolic calculation were determined using data from the two intense ride REHIT sessions. 

### 2.10. Statistical Analyses

All analyses were performed using SPSS Version 25.0 (Chicago, IL, USA) and GraphPad Prism 7.0. (San Diego, CA, USA). Measures of centrality and spread are presented as mean ± SD. Primary outcome measures for the acute cardiovascular and metabolic responses to the REHIT portion of the study were the relative exercise intensity (% HRR and % VO_2_R), metabolic equivalents (METs), blood pressure, and energy expenditure (kcal/min). Primary outcome measures for the chronic cardiovascular and metabolic responses to REHIT and MICT were the change in cardiometabolic risk factors, including CRF, weight, waist circumference, body composition, blood lipids, blood pressure, and blood glucose, and the MetS z-score. A two-way repeated measures analysis of covariance (ANCOVA) was used to examine the change in cardiometabolic factors between training groups, with the baseline value of each factor entered as the covariate and the post-intervention value as the dependent variable. To determine individual MetS z-score training responsiveness, delta values (Δ) were calculated (post-program minus baseline value) to establish the change (Δ) in the MetS z-score. The Δs in MetS z-score were compared to our calculated laboratory-specific coefficient of variability (CV) value of 0.60 [16]. Thus, for a participant to be considered a responder to overall MetS severity improvement, his/her MetS z-score change should be favorably greater than the established MetS z-score CV. Subsequently, participants were categorized as ‘1’ = responder if their Δ was greater than the laboratory-specific biological variability criterion or ‘0’ = non-responder if the Δ failed to exceed the laboratory-specific criterion. Chi-square (χ2) tests were used to analyze the incidence of responders and non-responders for the MetS z-score following the intervention separated by the treatment group (MICT and REHIT). The probability of making a Type I error was set at *p* < 0.05 for all statistical analyses. 

## 3. Results

All analyzes and data presented in the results are for those participants who completed the investigation. At baseline, treatment (REHIT and MICT) groups did not differ significantly in physical or physiological characteristics. The physical and physiological characteristics for participants in both groups at baseline and 8 wks are shown in Table 1. The exercise prescriptions in both treatment groups were well-tolerated for 27 of the 32 participants who completed the study. Five participants were unable to complete the study for the following reasons: Illness (n = 4) and personal reasons (n = 1). Dropout was greater in the REHIT group (n = 4) relative to the MICT group (n = 1); however, these differences can be explained by more illness in the REHIT group (n = 3) vs the MICT group (n = 1). Overall, there was excellent adherence to the total number of prescribed training sessions: REHIT group—mean, 89.2% (range, 75%–100%) and MICT group—mean, 87.8% (range, 77.1%–100%).

Changes in CRF (REHIT, 12%; MICT, 7%; F[1,24] = 9.9, *p* < 0.01), systolic blood pressure (REHIT, −5%; MICT, −2%; F[1,24] = 8.1, *p* < 0.01), waist circumference (REHIT, −1.4%; MICT, −0.3%; F[1,24] = 7.132, *p* = 0.01), and MetS severity (MetS z-score: REHIT, −62%; MICT, 27%; F[1,24] = 5.9, *p* = 0.02) were more favorable in the REHIT group when compared to the MICT group (Table 1) following the 8-wk program. There were significant within-group improvements (*p* < 0.05) in CRF, HDL cholesterol, triglycerides, and the MetS z-score in both groups (MICT and REHIT). However, only the REHIT group showed significant reductions (*p* < 0.05) in systolic blood pressure and waist circumference. 

### 3.1. Acute Cardiovascular and Metabolic Responses to REHIT

Acute cardiovascular and metabolic responses (range and mean ± SD) to the overall REHIT workout and the intense ride 2 × 20 s sprint sessions are presented in Table 2. Peak heart rate for the REHIT workouts were 168 beats/min, which corresponded to a 94% HRR. Peak metabolic equivalent (METs) was 8.8, which equated to a 92.9% oxygen uptake reserve (VO_2_Reserve). The peak systolic and diastolic blood pressure during the REHIT sessions equated to 218 and 98 mmHg, respectively.

### 3.2. Incidence of MetS z-Score Responders and Non-Responders

The incidence of MetS z-score responders and non-responders to exercise training in both the MICT and REHIT groups are shown in Figure 2. In the MICT group, 47% (7/15) of individuals experienced a favorable change in the MetS z-score (Δ > −0.60) and were categorized as responders. In the REHIT group, the incidence of individuals who experienced a favorable change in the MetS z-score were significantly (*p* < 0.05) greater when compared to the MICT group. Indeed, there was a positive improvement in the MetS z-score (Δ > −0.60) in 75% (9/12) of the individuals in the REHIT group. 

## 4. Discussion

The main finding of the present study was that 8 wks of REHIT elicited more potent and time-efficient improvements in CRF and cardiometabolic health when compared to traditional MICT. Indeed, there was nearly a two-fold greater improvement in CRF in the REHIT group (12.3%) relative to the MICT group (6.9%). Additionally, reductions in systolic blood pressure in the REHIT group were three-fold greater when compared to the MICT group (Table 1) and underpinned a ~4% reduction in mean arterial pressure (MAP). Another novel finding from the present study was that REHIT elicited a nearly 1.5-fold greater incidence of cardiometabolic responders (75%) as quantified by the MetS z-score when compared to the MICT group (47%). Given that a ‘lack of time’ is the most often cited reason for not exercising regularly, this study provides preliminary evidence that extends the application of the SIT paradigm from the scientific literature into a real-world workplace setting through the CAR.O.L ™ units which are developed for self-guided and public implementation of the REHIT protocol. 

Consistent with previous REHIT studies with a similar program duration (6–10 weeks) including sedentary individuals [11,17], our REHIT protocol also induced a ~12% increase in CRF. This is a clinically significant finding given that a 10% increase in CRF significantly reduces the risk of mortality and morbidity by 15% [18]. Moreover, congruent with a previous finding [19], we also showed that this CRF improvement following REHIT is superior to that elicited by a five-fold larger volume MICT. In contrast, Gillen et al. [20] reported a similar CRF increase following both REHIT and MICT in sedentary individuals. The discrepancies between these results may be due to the difference in the exercise program duration. Both the present study and Ruffino et al. [19] had the same 8-week program duration, whereas participants in Gillen et al.’s study [20] only performed 6 weeks of REHIT, suggesting that a minimum of 8 weeks of REHIT is required to surpass the beneficial effect of the traditional MICT on CRF. 

The specific mechanism by which REHIT can improve CRF more than MICT remains elusive. There is, however, evidence to suggest that the capacity of interval training to improve CRF more than MICT may be due to its ability to induce both central (i.e., increase in stroke volume) and peripheral adaptations (i.e., mitochondrial content/enzyme activity, and capillary density) [21]. Interestingly, Gillen et al. [20] showed no significant difference in mitochondrial content change following 6 weeks of REHIT and MICT, whereas Matsuo et al. [22] revealed significant changes in stroke volume, left ventricular mass, and resting heart rate following 8 weeks of interval training (SIT/HIIT), but not MICT. Collectively, these studies support our aforementioned hypothesis that at least 8 weeks of REHIT may be required to exceed the favorable CRF change following MICT.

The superior CRF improvement following REHIT relative to MICT in the present study was also accompanied by a similar pattern in cardiometabolic risk change, depicted as a reduction in MAP and overall MetS severity. Our MAP result is comparable with previous findings also showing a 4%–7% MAP reduction following REHIT [17,19]. This is a clinically significant finding as MAP is a major independent predictor of stroke [23]. This significant decrease in MAP could partly explain the superior reduction in overall MetS severity following REHIT compared to MICT, given that MAP contributes towards the calculation of MetS severity via the MetS z-score. Moreover, the superior ability of REHIT to improve MetS severity relative to MICT is not at all surprising given that CRF has repeatedly been reported to have an inverse relationship with MetS [24]. Earnest et al. [24] reported a significantly lower MetS z-score (MetS severity) in ‘high fit’ individuals compared to less fit individuals (‘low fit’ and ‘moderate fit’). This inverse relationship may be due to the significant positive association of CRF with insulin sensitivity [25] and vascular function [26], which are both mechanisms contributing to the severity of MetS [27,28]. Interval training has been reported to be more effective in improving insulin sensitivity [7] and vascular function [6] when compared to MICT. Interestingly, a previous study that utilized a similar REHIT protocol as the present study demonstrated improvements of insulin sensitivity in sedentary individuals to a similar extent as MICT [20]. This suggests that the superior improvement in MetS severity that was evident following REHIT relative to MICT could primarily be attributed to improved vascular function as reflected by a reduction in MAP. Further studies are warranted to test this hypothesis.

Although interval training has consistently been shown to confer health and fitness benefits [6,7], it is often criticized to impose unpleasant sensations in previously sedentary middle-aged men and women, ultimately affecting exercise adherence and compliance [29]. However, it has also been argued that this initial appraisal is based on the traditional definition of interval training, comprising of longer exposure to vigorous or high-intensity bouts of exercise compared to more practical and scalable protocols, such as REHIT, that have been continued to be developed as reflected in the present study. Indeed, previous interval protocols shown to be effective in improving cardiometabolic health usually require at least 1–4 min exercise bouts at vigorous to high-intensity that are repeated at least 4–10 times within a session, and essentially demand a comparable weekly time commitment as the current guideline (MICT) of 150 min per week [13]. In contrast, the REHIT protocol utilized in the present study only demanded one fifth of the time commitment required by MICT, yet demonstrated superior health benefits. Indeed, our study extends the findings of others reporting good adherence to REHIT sessions [12], suggesting that REHIT may be a more practical model than previous interval training protocols and the current guideline in the prevention of cardiometabolic diseases in time-poor middle-aged sedentary individuals. 

Safety is also a paramount issue when designing and implementing a HIIT program. Overall, it has been demonstrated that exercise is safe for most individuals, and exercise alone does not incite adverse cardiovascular or other untoward events [13]. Nevertheless, it has been shown that the risk of an acute myocardial infarction or sudden death during exercise is higher in adults compared to their younger counterpart; and that the greatest risk exists for those individuals with underlying or diagnosed CVD. Overall, the absolute risk of sudden death during vigorous-intensity, physical activity has been estimated to be one per year for every 15,000–18,000 people [13]. In a systematic review focused on the safety of HIIT for patients with CVD, it was found that there were no adverse cardiac or other life threatening events secondary to the prescription of HIIT [30]. Our REHIT results were consistent with these findings. Nevertheless, a key to minimizing complications during and after exercise is to identify those participants who may be at an increased risk of adverse symptoms through appropriate pre-participation screening. Participants with a clinically relevant CVD risk factor or diagnosed CVD may benefit from a medical examination and physician supervised maximal exercise test prior to participation in vigorous-intensity exercise [13]. Despite the absence of untoward events in the present study, given the near-maximal cardiovascular strain and hypertensive blood pressure responses to REHIT, these authors recommend careful pre-participation screening prior to the commencement of REHIT for previously inactive and/or risk factor burden individuals.

### Limitations

There are a few limitations to the present study that warrant further discussion. First, while participants were instructed to maintain their regular dietary intake during the 8 wk intervention, diet intake was not strictly controlled for in this study. Moreover, physical activity/sedentary behaviour outside of the training program and prescribed medications were not monitored, and thus may have influenced the current findings. Dropout was greater in the REHIT group (n = 4) when compared to the MICT group (n = 1); however, these differences appear to be explained by more illness in the REHIT group. Nevertheless, future research is needed to better ascertain whether REHIT is less-tolerable relative to MICT.

## 5. Conclusions

There is a wealth of existing research reporting that regular exercise training confers positive effects on CRF and numerous other cardiometabolic outcomes related to cardiovascular morbidity and mortality. Accordingly, substantial public health efforts have been aimed at promoting and increasing levels of physical activity. The main finding of the present study is that 8 wks of REHIT was safe and elicited more potent, time-efficient improvements in CRF and cardiometabolic health when compared to traditional MICT. Given that a ‘lack of time’ is the most often cited reason for not exercising regularly, this study provides critical evidence for ‘how’ to implement and translate the SIT paradigm into a real-world workplace setting. Future research is required in more diverse workplace settings beyond those examined in the present study. Additionally, the long-term adherence and effectiveness of REHIT requires scientific inquiry in order to determine its potential as a viable public health strategy. Overall, this knowledge may benefit researchers, along with exercise physiologists, health and fitness professionals, and others who design exercise programs and promote physical activity.

## Figures and Tables

**Figure 1 ijerph-16-00483-f001:**
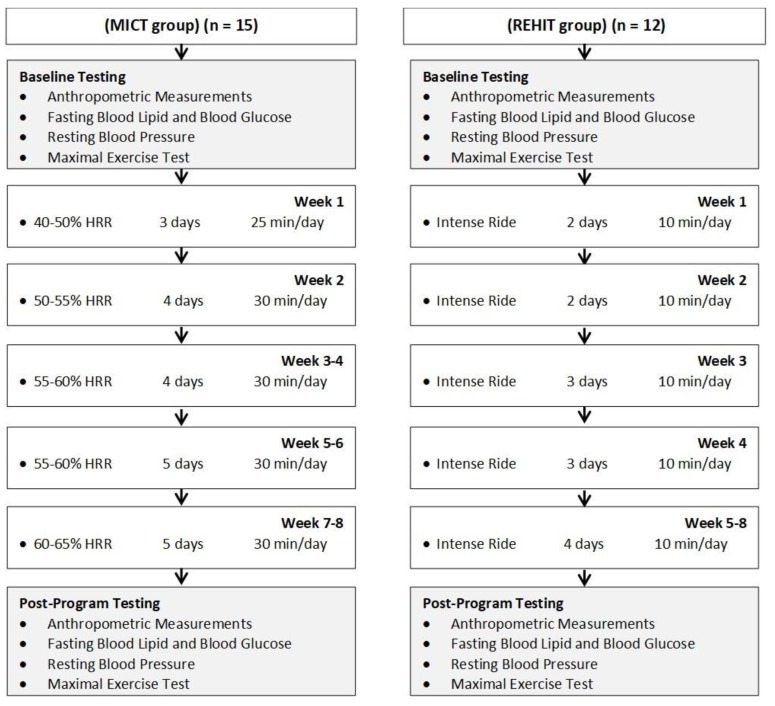
Experimental flow diagram and week-to-week exercise prescription for the days/times of training.

**Figure 2 ijerph-16-00483-f002:**
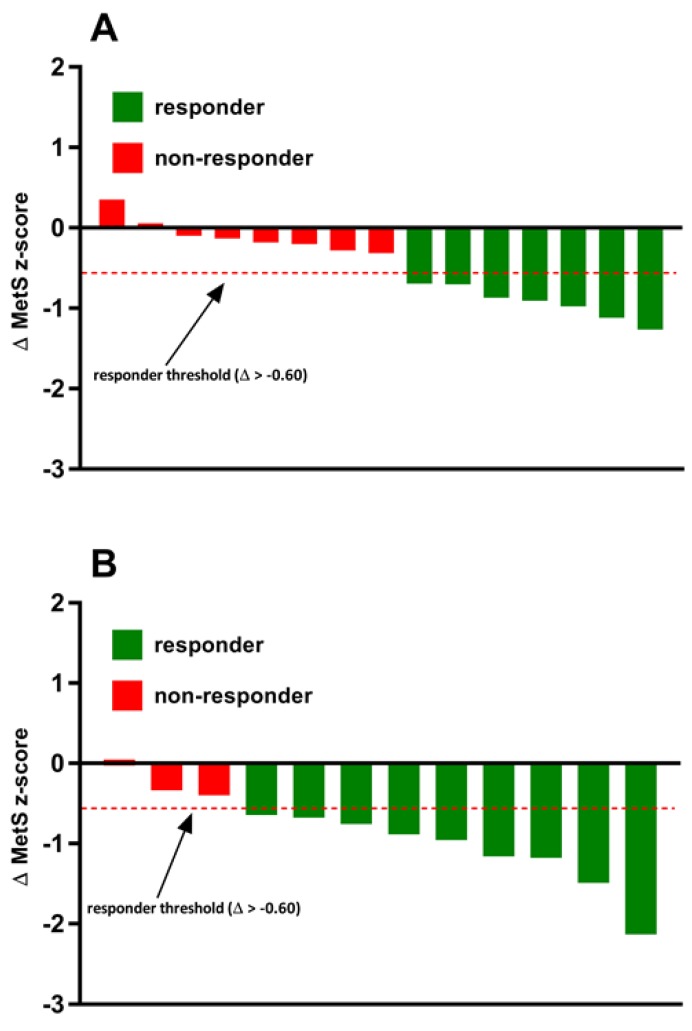
Inter-individual variability in the MetS z-score responses to exercise training in the (**A**) MICT group and (**B**) REHIT group.

**Table 1 ijerph-16-00483-t001:** Physical and physiological characteristics at baseline and 8 wks for REHIT and MICT groups.

Parameter	MICT Group (n = 15)		REHIT Group (n = 12)	
	Baseline	8 wk	Baseline	8 wk
Age (yr)	42.2 ± 9.7 ^a^	____	40.8 ± 10.8	____
Height (cm)	170.1 ± 10.9	____	172.8 ± 9.8	____
Weight (kg)	79.2 ± 14.4	78.7 ± 13.5	81.9 ± 11.3	82.4 ± 11.3
Waist circumference (cm)	87.6 ± 10.1	87.3 ± 9.1	86.6 ± 10.5	85.4 ± 9.7 *^,†^
Body fat (%)	31.1 ± 4.1	30.3 ± 4.2	32.8 ± 6.2	31.9 ± 4.7
CRF (mL·kg^−1^·min^−1^)	26.2 ± 6.7	28.0 ± 7.1 *	25.3 ± 2.9	28.4 ± 3.1 *^,†^
Systolic BP (mmHg)	128.9 ± 17.4	126.7 ± 14.4	130.4 ± 8.8	123.8 ± 6.6 *^,†^
Diastolic BP (mmHg)	82.5 ± 9.6	81.7 ± 7.1	83.2 ± 5.6	82.2 ± 4.9
HDL cholesterol (mg·dL^−1^)	46.6 ± 5.9	48.3 ± 6.3 *	44.8 ± 6.4	47.6 ± 5.1 *
Triglycerides (mg·dL^−1^)	131.1 ± 32.1	123.9 ± 30.1 *	138.8 ± 36.2	125.2 ± 24.0 *
Blood Glucose (mg·dL^−1^)	95.3 ± 6.8	94.1 ± 10.0	96.7 ± 6.7	94.9 ± 5.6
MetS z-score	−1.83 ± 2.24	−2.32 ± 2.12 *	−1.56 ± 1.49	−2.53 ± 1.29 *^,†^

^a^ Values are mean ± SD; * Within-group change is significantly different from baseline, *p* < 0.05; ^†^ Change from baseline is significantly different than the MICT group, *p* < 0.05.

**Table 2 ijerph-16-00483-t002:** Acute cardiovascular and metabolic responses to the REHIT workout.

Parameter	Range (Lowest)	Range (Highest)	Overall REHIT Workout	Intense Ride 2 × 20 s Sprints
HR (beats/min)	76	168	120.9 ± 13.8 ^a^	152.6 ^a^ ± 14.1
%HRR	7.2	94.0	44.2 ± 8.1	86.1 ± 4.4
%VO_2_R	8.7	92.9	33.7 ± 6.5	74.4 ± 11.5
METs	1.2	8.8	3.4 ± 0.9	6.2 ± 1.3
kcal/min	1.3	14.8	4.6 ± 2.1	8.2 ± 2.7
SBP (mmHg)	126	218	154.8 ± 31.8	160.9 ± 30.5
DBP (mmHg)	62	98	76.8 ± 11.8	78.0 ± 8.8

^a^ Values are mean ± SD; DBP, diastolic blood pressure; HR, heart rate; %HRR, percentage heart rate reserve; kcal, kilocalories; METs, metabolic equivalents; SBP, systolic blood pressure; %VO_2_R, percentage oxygen uptake reserve.

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
