# Peer review of "Reduced Exertion High-Intensity Interval Training is More Effective at Improving Cardiorespiratory Fitness and Cardiometabolic Health than Traditional Moderate-Intensity Continuous Training"

_ijerph, 2019, doi:10.3390/ijerph16030483_

Reviewer 1 Report

Spodná časť formulára

The study determines the effectiveness of 8wk reduced-exertion high-intensity interval training (REHIT) at improving cardiorespiratory fitness (CRF) and positively modifying cardiometabolic health in the workplace environment. The authors concluded that 8wk REHIT elicites more potent and time-efficient improvements in CRF and cardiometabolic health when compared to traditional MICT.

The results of this research are in my opinion of relevance to public health research and practice and would fit the scope of the journal. There are, however, some minor concerns which should be addressed.

The authors should report whether the sampling was sufficient. There is no information whether the power analysis for estimating appropriate sample size was provided.

Page 3, Table 1: MICT group (n = 14), REHIT group (n = 12)

Please, specify the number of women and men participated in each group.

Page 2, lines 74-76: A cohort of men and women (n=32) were recruited from the faculty/employee population of a local university and hospital via advertisement through the university website, local community newspaper, and word-of-mouth.

The discussion reflects what authors found and how it relates to the literature. The authors incorporated previous research into their interpretation of the results. They demonstrated an appropriate understanding of previous research on the topic and included related references. However, the discussion should also present the practical application of the obtained findings and recommendations with respect to specific population of middle-aged men and women with a predominantly sedentary lifestyle.

Author Response

IJREPH comments

Reviewer #1 comments:

The study determines the effectiveness of 8wk reduced-exertion high-intensity interval training (REHIT) at improving cardiorespiratory fitness (CRF) and positively modifying cardiometabolic health in the workplace environment. The authors concluded that 8wk REHIT elicites more potent and time-efficient improvements in CRF and cardiometabolic health when compared to traditional MICT.

The results of this research are in my opinion of relevance to public health research and practice and would fit the scope of the journal. There are, however, some minor concerns which should be addressed.

Comment 1. The authors should report whether the sampling was sufficient. There is no information whether the power analysis for estimating appropriate sample size was provided.

Thank you for this comment.

We used a comparable sample size to another one of our recently published studies in the International Journal of Environmental Research and Public Health:

Weatherwax, R.M.; Ramos, J.S.; Harris, N.K.; Kilding, A.E.; Dalleck, L.C. Changes in Metabolic Syndrome Severity Following Individualized Versus Standardized Exercise Prescription: A Feasibility Study. Int. J. Environ. Res. Public Health 2018, 15, pii: E2594.

Nevertheless, we did not specifically perform an apriori power calculation. However, given the significant differences between groups it is our contention that our statistical power was sufficient. Please advise if further details are required.

Comment 2. Page 3, Table 1: MICT group (n = 14), REHIT group (n = 12)

Thank you for this comment. We have corrected Figure 1 to align with what we have reported in the results section and Table 1. Table 1 is correct: MICT group (n=15), REHIT group (n=12).

Comment 3. Please, specify the number of women and men participated in each group.

Page 2, lines 74-76: A cohort of men and women (n=32) were recruited from the faculty/employee population of a local university and hospital via advertisement through the university website, local community newspaper, and word-of-mouth.

Thank you for this suggestion. We have added these details to the revised manuscript.

‘A cohort (n=32) of men (n=16) and women (n=16)…’

Comment 4. The discussion reflects what authors found and how it relates to the literature. The authors incorporated previous research into their interpretation of the results. They demonstrated an appropriate understanding of previous research on the topic and included related references. However, the discussion should also present the practical application of the obtained findings and recommendations with respect to specific population of middle-aged men and women with a predominantly sedentary lifestyle.

Thank you for this suggestion. We have added the following to the revised discussion.

‘Although interval training has consistently been shown to confer health and fitness benefits [6-7], it is often criticized to impose unpleasant sensations in previously sedentary middle-aged men and women, ultimately affecting exercise adherence and compliance [29]. However, it has also been argued that this initial appraisal is based on the traditional definition of interval training, comprising of longer exposure to vigorous or high-intensity bouts of exercise compared to more practical and scalable protocols such as REHIT that have been continued to be developed as reflected in the present study. Indeed, previous interval protocols shown to be effective in improving cardiometabolic health usually require at least 1-4 min exercise bouts at vigorous to high-intensity that are repeated at least 4-10 times within a session, and essentially demand comparable weekly time-commitment as the current guideline (MICT) of 150 min per week [13]. In contrast, the REHIT protocol utilized in the present study only demanded 1/5 of the time-commitment required by MICT, yet demonstrated superior health benefits. Indeed, our study extends the findings of others reporting good adherence to REHIT sessions [12], suggesting that REHIT may be a more practical model than previous interval training protocols and the current guideline in the prevention of cardiometabolic diseases in time-poor middle-aged sedentary individuals.’

Reviewer 2 Report

I found the research very significant and the need for studies in this area. I realize the number of subjects were low because of the type of research and then unforeseen circumstances of illness. I thought a little more could of been provided for the Limitation of results and conclusion but there was enough for the reader to understand the outcome and limitations of the study

Reduced exertion high-intensity interval training is more effective at improving cardiorespiratory fitness

and cardiometabolic health than traditional moderate-intensity continuous training 

Tom F. Cuddy 1, Joyce S. Ramos 2,3 and Lance C. Dalleck 1,2,*

Comments highlighting areas of the

Strengths

·       Choice of Research study

o   The purpose of the study to research to investigate high interval training for improving  cardiorespiratory fitness vs traditional moderate-intensity continuous training is very significant for  prevention of cardiorespiratory incidence in this day and age  and for future programs to encourage people for a healthy lifestyle that won’t disrupt their daily routines.

·       Study Design

o   The method and selection of subjects from the University and hospital employees was an ideal group to select to participate

o   The design of the study with 8 week commitment and a baseline with control and experimental group met research typical research design for this type of study

o   The measurements selected in the study were varied with blood levels (Fasting blood lipid and blood glucose measurement, Resting Blood Pressure measurement and adequate for the design Anthropometric measurement

·       Reporting of results

o    The use of Metabolic syndrome z-score, maximum exercise  and intensity exercise and metabolic calculations with statistical analysis was adequate for this type of student

o   The results were clearly analyzed for this type of study with adequate tables to demonstrate comparison within and between groups

·       Discussion

o   The discussion clearly gave insights into the design, method and results of the study and the issues that the  results were consistent with the findings

Weakness

·       Limitations

o   The limitation section could be expanded with any anecdotal analysis as to why subjects dropped out e.g. was it an usual flu season not anticipated, were their unexpected family issues, was the time and location of the exercise program an issue

·       Conclusion

o   The conclusion section could be expanded into needs for future studies, use of different subjects use in different work environments

o   In addition to supporting existing research in the area, how can the researchers springboard this study as a baseline for expanding their future research in the area and encourage other researchers to simulate their own research

Author Response

IJREPH comments

Reviewer #2 comments:

I found the research very significant and the need for studies in this area. I realize the number of subjects were low because of the type of research and then unforeseen circumstances of illness. I thought a little more could of been provided for the Limitation of results and conclusion but there was enough for the reader to understand the outcome and limitations of the study.

Thank you for the positive comments and suggestions. As you have suggested, we have made some additions to the limitations section and conclusion. 

Reduced exertion high-intensity interval training is more effective at improving cardiorespiratory fitness

and cardiometabolic health than traditional moderate-intensity continuous training 

Tom F. Cuddy 1, Joyce S. Ramos 2,3 and Lance C. Dalleck 1,2,*

Comments highlighting areas of the

Strengths

·   The purpose of the study to research to investigate high interval training for improving  cardiorespiratory fitness vs traditional moderate-intensity continuous training is very significant for  prevention of cardiorespiratory incidence in this day and age  and for future programs to encourage people for a healthy lifestyle that won’t disrupt their daily routines.

·   The method and selection of subjects from the University and hospital employees was an ideal group to select to participate

o   The measurements selected in the study were varied with blood levels (Fasting blood lipid and blood glucose measurement, Resting Blood Pressure measurement and adequate for the design Anthropometric measurement

·    The use of Metabolic syndrome z-score, maximum exercise  and intensity exercise and metabolic calculations with statistical analysis was adequate for this type of student

o       Discussion

o       Limitations

The limitation section could be expanded with any anecdotal analysis as to why subjects dropped out e.g. was it an usual flu season not anticipated, were their unexpected family issues, was the time and location of the exercise program an issue

Thank you – the following was added to the revised limitations section:

Dropout was greater in the REHIT group (n=4) when compared to the MICT group (n=1); however, these differences appear to be explained by more illness in the REHIT group. Nevertheless, future research is needed to better ascertain whether REHIT is less-tolerable relative to MICT.

Conclusion

The conclusion section could be expanded into needs for future studies, use of different subjects use in different work environments

In addition to supporting existing research in the area, how can the researchers springboard this study as a baseline for expanding their future research in the area and encourage other researchers to simulate their own research.

Thank you – the following was added to the revised conclusion section:

Future research is required in more diverse workplace settings beyond those examined in the present study. Additionally, the long-term adherence and effectiveness of REHIT requires scientific inquiry in order to determine its potential as a viable public health strategy.
